# Renewable Power and Heat for the Decarbonisation of Energy-Intensive Industries

**Alessandro A. Carmona-Martínez** [1],***, **Alejandro Fresneda-Cruz** [1], **Asier Rueda** [1], **Olgu Birgi** [2], **Cosette Khawaja** [2], **Rainer Janssen** [2], **Bas Davidis** [3], **Patrick Reumerman** [3], **Martijn Vis** [3], **Emmanouil Karampinis** [4], **Panagiotis Grammelis** [4] and **Clara Jarauta-Córdoba** [1]

1   CIRCE—Research Centre for Energy Resources and Consumption, Parque Empresarial Dinamiza, Ave. Ranillas 3D, 1st Floor, 50018 Zaragoza, Spain
2   WIP—WIP Renewable Energies, Sylvensteinstraße 2, 81369 Munich, Germany
3   BTG—BTG Biomass Technology Group, Josink Esweg 34, 7543 ES Enschede, The Netherlands
4   CERTH—Centre for Research and Technology Hellas/Chemical Process and Energy Resources Institute, Egialias 52, 15125 Marousi, Greece
*   Correspondence: acarmona@fcirce.es

**Abstract:** The present review provides a catalogue of relevant renewable energy (RE) technologies currently available (regarding the 2030 scope) and to be available in the transition towards 2050 for the decarbonisation of Energy Intensive Industries (EIIs). RE solutions have been classified into technologies based on the use of renewable electricity and those used to produce heat for multiple industrial processes. Electrification will be key thanks to the gradual decrease in renewable power prices and the conversion of natural-gas-dependent processes. Industrial processes that are not eligible for electrification will still need a form of renewable heat. Among them, the following have been identified: concentrating solar power, heat pumps, and geothermal energy. These can supply a broad range of needed temperatures. Biomass will be a key element not only in the decarbonisation of conventional combustion systems but also as a biofuel feedstock. Biomethane and green hydrogen are considered essential. Biomethane can allow a straightforward transition from fossil-based natural gas to renewable gas. Green hydrogen production technologies will be required to increase their maturity and availability in Europe (EU). EIIs' decarbonisation will occur through the progressive use of an energy mix that allows EU industrial sectors to remain competitive on a global scale. Each industrial sector will require specific renewable energy solutions, especially the top greenhouse gas-emitting industries. This analysis has also been conceived as a starting point for discussions with potential decision makers to facilitate a more rapid transition of EIIs to full decarbonisation.

**Keywords:** energy-intensive industries; decarbonisation; renewable energies; biomass; green hydrogen; heat pumps; solar thermal; geothermal

## 1. Introduction

The EU has set ambitious targets for decarbonisation by 2050 [1]. Part of this decarbonisation relies on the implementation of renewable energy (RE) technologies that replace the use of fossil-based energies. The second pillar of this decarbonisation path must be built on avoiding the emission of greenhouse gases (GHG) into the atmosphere by energy-intensive industries (EIIs). It is well known that EIIs were responsible for a third of the total of such emissions (>508 Mt CO2e) in the EU in 2014 [2,3]. In 2020, the European Union consumed around 2685 TWh of energy. The chemical and petrochemical sectors had the highest energy consumption (22%), followed by the non-metallic mineral (14%); paper, pulp, and printing (14%); and food, beverage, and tobacco (12%) sectors. These four industrial sectors alone consumed more than 60% of the total energy used by European industries (Figure 1) [4].

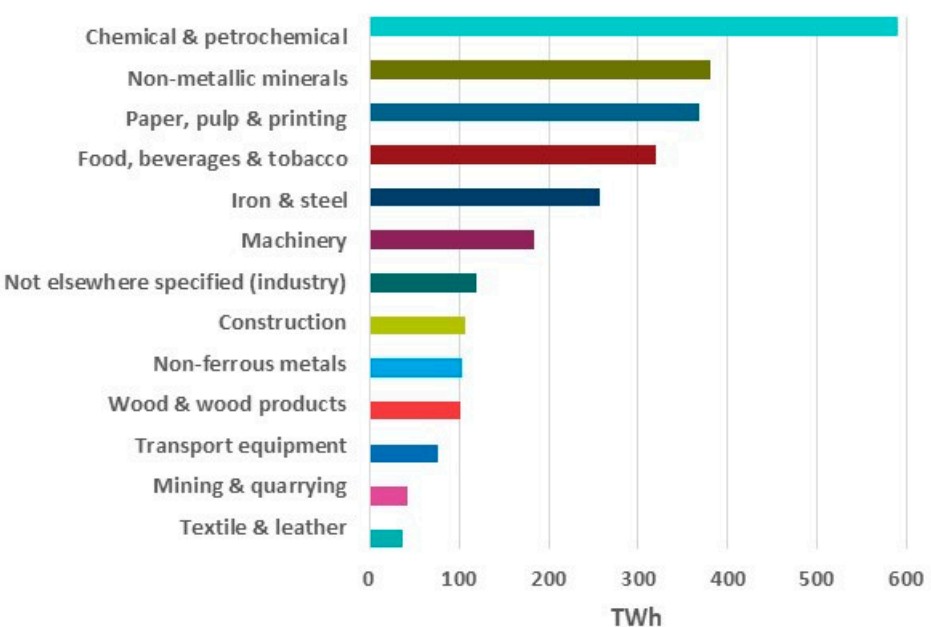

**Figure 1.** European energy consumption in 2020 by industrial sector. Plotted from information in the Eurostat database [4].

The present work provides an assessment of the most relevant technologies for renewable electricity and heat production as a replacement for fossil-based energy consumption by EIIs. Relevant RE technologies that can be deployed either in the short- (i.e., until 2030) or long-term (until 2050) have been analysed, as well as their integration into EIIs and in combination with other RE technologies. Within the 2030 scope, the analysis of relevant renewables based on their capacity to produce clean energy has been divided in the following two categories:

i. Replacement of existing fossil fuel-based electricity with clean renewable electricity sources such as wind, solar photovoltaic, and hydropower. This is a well-known group of relevant renewable technologies that are implemented in industries mainly through renewable Power Purchase Agreements (PPAs).

ii. Replacement of existing fossil fuel-based heat-produced heat with renewable heat production technologies such as solar thermal, heat pumps, geothermal energy, green hydrogen, and bioenergy such as solid biomass and liquid and gaseous biofuels. Currently, these REs show limited implementation in EIIs, even though they have been identified as a promising route for the decarbonisation of the sector.

Both groups of RE technologies considered in this analysis, renewable electricity and renewable heat, are being assessed on the basis of their technical feasibility, firstly by taking into consideration their availability by 2030. Secondly, the considered renewable solutions are being analysed in terms of what is required for their implementation. Renewable energy solutions for EIIs for the 2050 horizon are also being analysed. Some have already been identified, including e-fuels or renewable synthetic fuels from, e.g., the hydrogenation of captured $CO_2$, among others.

## 2. Renewable Electricity

*Solar Photovoltaic, Concentrating Solar Power, and On/Offshore Wind*

Renewable power can be obtained from different sources: solar photovoltaic, concentrating solar power, and on/offshore wind. The last decade has seen price improvements in these technologies for power production. According to a recent analysis by IRENA [5], the price of these sources of renewable power has steadily decreased (Figure 2). In comparison, fossil-based energy sources such as coal-fired power plants have operating costs that are higher than their renewable counterparts. IRENA's analysis—focused on Europe, North

America, and South Asia—indicates that the costs of these renewables vary, in part due to the price imposed on $CO_2$ emissions. Figure 2 indicates that renewable technology power production prices have experienced a considerable decline since 2010. This shows the competitiveness of renewable power generation and that the electrification process required for the decarbonisation of energy-intensive industries could experience a similar cost-declining trend.

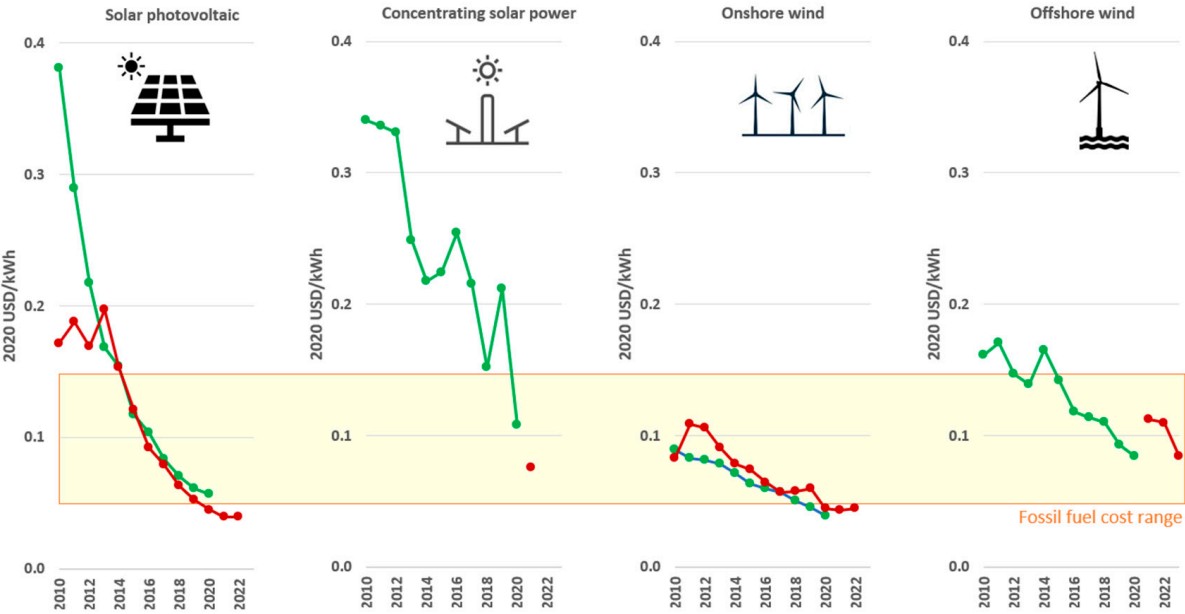

**Figure 2.** Global weighted-average levelised costs of energy (green) and power purchase agreement auction (red) prices for solar photovoltaic, on/offshore wind, and concentrating solar power between 2010 and 2023. Plotted from the original source [5].

The main advantage of renewable power is its flexibility in terms of implementation. Grid-connected installations harvest electricity for self-consumption and the surplus can be given to the network. However, off-grid facilities operate in isolation. These are placed in remote locations to meet local electricity demands. Off-grid facilities require the installation of batteries to store surplus electricity. By 2021, the European Union had installed around 26.8 GW of photovoltaic capacity [6]. The largest European market was Germany (21%), followed by Spain (19%), France (14%), the Netherlands (13%), Poland (13%), Greece (4%), and Italy (4%). The current worldwide photovoltaic power capacity is expected to grow from 900 GW (EU share of 25%) to 3000 GW (EU share of 5%) by 2050 according to the IEA Roadmap [7].

There are two main pathways for the implementation of renewable electricity produced either from solar photovoltaic, concentrating solar power, or on/offshore wind. The most straightforward pathway is the direct substitution of fossil-based electricity in current industrial processes. The second pathway involves the electrification of current processes based on a heat supply obtained from the use of non-renewable fuels such as natural gas and coal, among others [8].

Electrically powered technologies cover the broad temperature spectrum required by industries [9]. Applications that require low and medium temperatures, such as electric boilers and heat pumps that supply heat and cooling, are not sector-specific and can thus be implemented transversally. Table 1 presents a portfolio of electrically powered technologies expected to substitute conventional fossil-based energy (either non-renewable electricity or gas) for renewable heating and cooling.

**Table 1.** Renewable electricity-based technologies for the electrification of industrial processes.

| Process Temperature Range in °C | | | | Technological Maturity | Applications | Electrification Stages |
|---|---|---|---|---|---|---|
| **<100** | **100–400** | **400–1000** | **>1000** | | | |
| Compression heat pumps and chillers | | N.A. | N.A. | Established in industries (only <100 °C) | Space heating<br>Hot water<br>Low-pressure steam<br>Drying<br>Cooling and refrigeration | 1 |
| Mechanical vapour recompression (MVR) | | N.A. | N.A. | Established in industries | Energy recovery<br>Distillation<br>Evaporation<br>Steam generation<br>Process heat | 1 |
| Electric boilers | | | N.A. | Established in industries | Space heating<br>Hot water<br>Thermal oil<br>Steam | 1 |
| Infrared heaters | | | | Established in industries | Drying<br>Paint curing<br>Plastic treatments<br>Food processing | 1 |
| Microwave and radio-frequency heaters | | | | Established in industries, except for cement and ceramic firing/sintering | Drying<br>Ceramic firing<br>Ceramic sintering<br>Cement treatment<br>Food processing | 1 |
| Induction furnaces | | | | Established in industries | Metal melting<br>Reheating<br>Annealing<br>Welding | 1 |
| Resistance furnaces | | | | Established in industries | Metal melting<br>Smelting<br>Chemical heating<br>Ceramic firing<br>Glass melting<br>Calcination | 2, 3 |
| Electric arc furnaces | | | | Established in industries | Metal melting<br>Partial refining | 2, 3 |
| Plasma technology | | | | Established in industries only for metal and waste treatment | Waste treatment<br>Metal treatment<br>Welding<br>Sintering<br>Cement production | 2, 3 |

Notes: Electrification stages: (1) Includes thermal processes that are common to all industries and therefore considered potential entry points for electrification; (2) corresponds to a more advanced stage of electrification with sector-specific technologies; and (3) explores the maximum achievable electrification potential when considering technologies with higher uncertainties and lower technological readiness levels. Table modified from the original source [9].

## 3. Renewable Heat

### 3.1. Solar Thermal

Solar thermal heat is the energy produced by converting solar energy into usable heat. Solar thermal collectors are the devices used for this purpose. They absorb the incoming solar radiation, convert it into heat, and transfer this heat to a medium (usually air, water,

or oil) flowing through the collector. The solar energy collected is transported by the circulating fluid to be used directly or stored in a thermal energy storage tank [10].

There are two types of solar collectors: non-concentrating or stationary and concentrating. A non-concentrating collector has the same area for intercepting and absorbing solar radiation, whereas a sun-tracking concentrating solar collector usually has concave reflecting surfaces to intercept and focus the sun's radiation on a smaller receiving area, thereby increasing the radiation flux [11]. Table 2 shows the most common types of collectors and the temperature ranges that they can deliver.

**Table 2.** Common types of collectors and the temperature ranges they can deliver.

| Motion | Collector Type | Absorber Type | Temperature (°C) |
|---|---|---|---|
| Stationary non-concentrating | Flat plate collector (FPC) | Flat | 30–80 [12] |
| | Evacuated tube collector (ETC) | Flat | 50–200 [12] |
| Concentrating (single-axis tracking) | Parabolic trough collector (PTC) | Tubular | 60–375 [13] |
| | Linear Fresnel collector (LFC) | Tubular | 60–400 [14] |
| Concentrating (two-axis tracking) | Parabolic dish collector (PDC) | Point | 750–1000 [15] |
| | Power tower receiver | Point | 500–1500 [16] |

Non-concentrating solar heaters are already in use on a commercial scale. Parabolic troughs have also reached commercial maturity, with well-documented references concerning their availability and reliability [13]. Linear Fresnel collectors are less mature than troughs, but they are also available on a commercial scale [17]. Parabolic dish and power tower receivers also exist on a commercial scale but they are still at the initial stage of commercialisation in Europe. The total overall production of solar thermal energy in the EU-28 countries in 2016 was around 50 TWh, representing a 2% share of renewable energy [18].

The solar process heat installations applied to industrial sectors are similar to those used in residential buildings, especially for those applications where low (<150 °C) to medium (150 °C–400 °C) temperatures are required. For higher temperatures (>400 °C), more advanced or concentrated solar collectors are required. Almost all industrial processes with a heat demand require temperatures that can be provided by a solar thermal system. Among the EIIs, the chemical sector has a high percentage of low- and medium-temperature heat requirements in its production processes (>50%) and is the most suitable industrial sector (among the EIIs) for the effective use of solar thermal heat. The selection of an appropriate solar collector depends on several factors including operating temperature, thermal efficiency, energy yield, and costs, among others [19].

### 3.2. Heat Pumps

A heat pump (HP) is a technology that provides heating, cooling, and hot water. There are multiple known applications of heat pumps focused on district heating [20,21]. However, the use of heat pumps for industrial applications is gaining interest due to their potential to aid in the decarbonisation of processes. Heat pumps convert energy from air, ground, and water to useful heat using a refrigerant cycle. Such a cycle runs thanks to a special fluid that undergoes phase transitions and circulates in a closed circuit, which is normally composed of four parts: an evaporator, compressor, condenser, and expansion valve [22].

Air-source heat pumps use the heat of the surrounding ambient air as their primary energy source. These types of heat pumps are considered less expensive compared to other existing heat-pump-based technologies. Air-source heat pumps are installed above ground. Another type of air-source heat pump applied to industrial processes is the so-called exhaust heat pump. It uses the exhaust heat from manufacturing processes. Exhaust heat is typically warmer than the surrounding air. This causes the evaporation-to-condensation process to be more effective. These types of heat pumps are very suitable for the industrial sector due to the available heat streams that could potentially be recovered.

Underwater heat pumps require a water source as the heat-exchange medium, which could be obtained either from the ground, surface, or seawater [23]. These types of heat pumps extract heat from the water source, making the heat available for other applications such as heating, cooling, and the preparation of hot water. Water heat pumps are considered highly efficient due to the excellent temperature characteristics of water as an energy carrier. Additionally, the underwater temperature remains stable throughout the year. This type of heat pump is especially interesting for locations where extreme weather does not cause a drop in performance.

The implementation of heat pumps in EIIs is limited for multiple reasons. There are not enough manufacturers of equipment based on the concept of heat pumps. The available equipment is not able to deliver the broad range of process temperatures typically required by industries. Most commercial manufacturers provide equipment able to supply heat with temperatures up to 90 °C. Above such temperatures, the available options in the market are constrained. Only a few providers offer equipment that is able to deliver heat in the range of 120 to 165 °C. Fortunately, several ongoing projects have demonstrated heat delivery in the range of 160 to 200 °C [24]. EIIs have a large heat demand, requiring temperatures up to 200 °C. Table 3 provides an overview of the potential of heat-pump applications in several industrial sectors [25].

**Table 3.** Overview of heat-pump applications and temperature ranges in selected industrial processes.

| Industrial Sector | Industrial Process | Process Temperature in °C | | | | | | | | | | T in °C |
|---|---|---|---|---|---|---|---|---|---|---|---|---|
| | | 20 | 40 | 60 | 80 | 100 | 120 | 140 | 160 | 180 | 200 | |
| Paper | Drying | | | | | | | | | | | 90–240 |
| | Boiling | | | | | | | | | | | 110–180 |
| | Bleaching | | | | | | | | | | | 40–150 |
| | De-inking | | | | | | | | | | | 50–70 |
| Food and Beverage | Drying | | | | | | | | | | | 40–250 |
| | Evaporation | | | | | | | | | | | 40–170 |
| | Pasteurisation | | | | | | | | | | | 60–150 |
| | Sterilisation | | | | | | | | | | | 100–140 |
| | Boiling | | | | | | | | | | | 70–120 |
| | Distillation | | | | | | | | | | | 40–100 |
| | Blanching | | | | | | | | | | | 60–90 |
| | Scalding | | | | | | | | | | | 50–90 |
| | Concentration | | | | | | | | | | | 60–80 |
| | Tempering | | | | | | | | | | | 40–80 |
| | Smoking | | | | | | | | | | | 20–80 |

Table 3. *Cont.*

| Industrial Sector | Industrial Process | Process Temperature in °C | T in °C |
|---|---|---|---|
| | | 20  40  60  80  100  120  140  160  180  200 | |
| Chemicals | Distillation | | 100–300 |
| | Compression | | 110–170 |
| | Thermoforming | | 130–160 |
| | Concentration | | 120–140 |
| | Boiling | | 80–110 |
| | Bioreactions | | 20–60 |
| Metals | Drying | | 60–200 |
| | Pickling | | 20–100 |
| | Degreasing | | 20–100 |
| | Electroplating | | 30–90 |
| | Phosphating | | 30–90 |
| | Chromating | | 20–80 |
| | Purging | | 40–70 |
| Plastic | Injection moulding | | 90–300 |
| | Pellet drying | | 40–150 |
| | Preheating | | 50–70 |
| Textiles | Colouring | | 40–160 |
| | Drying | | 60–130 |
| | Washing | | 40–110 |
| | Bleaching | | 40–110 |
| Wood processing | Glueing | | 120–180 |
| | Pressing | | 120–170 |
| | Drying | | 40–150 |
| | Steaming | | 70–100 |
| | Cocking | | 80–90 |
| | Staining | | 50–80 |
| | Pickling | | 40–70 |
| Automotive | Resin moulding | | 70–130 |
| Mechanical engineering | Surface treatment | | 20–120 |
| | Cleaning | | 40–90 |

Technology Readiness Level (TRL) indicated above:
- Conventional HP (<80 °C), established in industries
- Commercially available HP (80 to 100 °C), key technology
- Prototype status, high-temperature HP (100–140 °C)
- Laboratory research status, very-high-temperature HP (>140 °C)

Table modified from the original source [25,26].

### 3.3. Geothermal

Geothermal energy is increasingly seen as an energy source that will aid in the decarbonisation of European industries. There are projections that by 2050, around 100 to 210 TWh/year will be available using geothermal energy [27]. The main applications of geothermal energy have been in the residential and commercial sectors in the form of district heating [28,29]. However, applications in the agricultural and industrial sectors are

also expected [30]. In industries, geothermal energy can be directly employed to supply heat or steam for processes as diverse as pasteurisation, drying, and evaporation, among others. Geothermal heat and steam could be implemented in industries such as the food-processing industry, chemical production, and material mining. Additionally, the benefits of geothermal energy sources include the provision of local, flexible renewable energy; diversification of the energy mix; reduction in fossil-based fuel imports; and protection against volatile fossil fuel prices [31].

Geothermal energy applications are highly dependent on the below-surface water temperature. According to Dalla Longa and colleagues [31], practically all regions in Europe show an economical potential for geothermal energy applications depending on the depth (Figure 3). Except for Iceland and a few other European regions with clear volcanic activity, the potential to produce electricity from geothermal energy is limited to reservoirs of depths less than 2 km. Direct geothermal applications, such as in agricultural greenhouses or industries, can be developed when reservoirs with depths of less than 2 km are available.

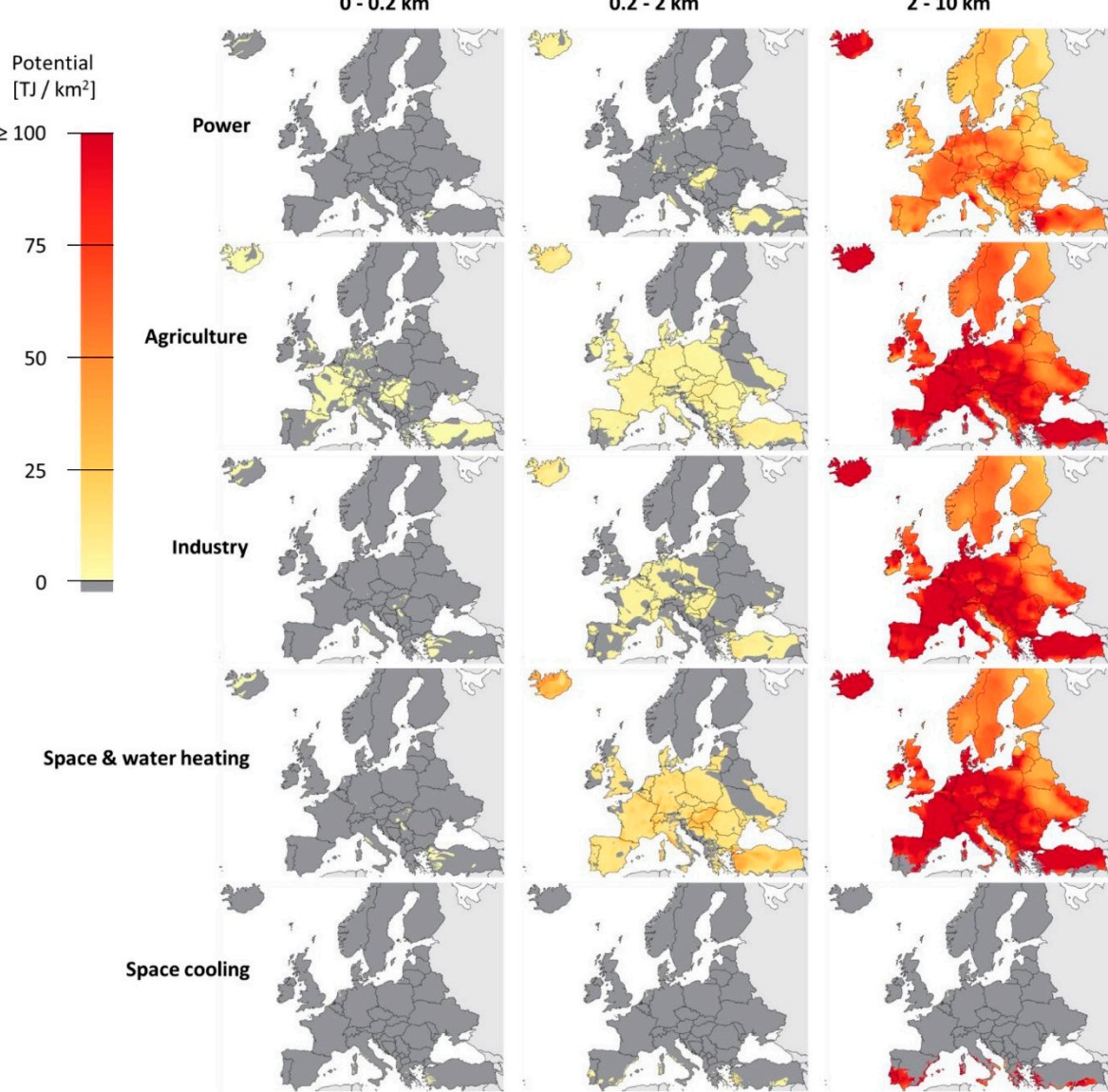

**Figure 3.** Long-term economic potential for various geothermal applications in Europe at three different ranges. Taken from the original source [28] (license: CC by 4.0).

Historically, European countries that have taken advantage of geothermal energy, such as Iceland, Italy, France, and Hungary, among others, have been the first to develop applications based on this energy source. However, applications based on the use of geothermal energy can also be developed in other low- and medium-enthalpy areas [32]. In regions with lower-temperature geothermal sites, applications make use of heat pumps [31].

One challenge concerning the use of geothermal-based energy is related to the financing and development of the infrastructure for a new heat grid [33]. Retrofitting is seen as an alternative to the implementation of geothermal energy not only for its most common application—urban district heating—but also as an energy source for energy-intensive industries.

### 3.4. Solid Biomass

Solid biomass has been identified as a key fuel for the transition to renewable energies in Europe. It is by far the main feedstock (91%) for bioheat production [34]. There are many conversion processes needed to transform biomass into useful forms of energy, which can be categorised into three main conversion pathways: thermochemical, physicochemical, and biochemical. Renewable heat can be produced using technologies that are characterised as thermochemical conversion processes.

Figure 4 illustrates the main thermochemical conversion technologies able to produce renewable heat and power from solid biomass. One of the main advantages of these technologies is their versatility, which makes it possible to take advantage of a wide variety of raw materials that can be used as fuels. Another advantage is that the energy generated is non-intermittent, which means that the required quantities can be generated when needed.

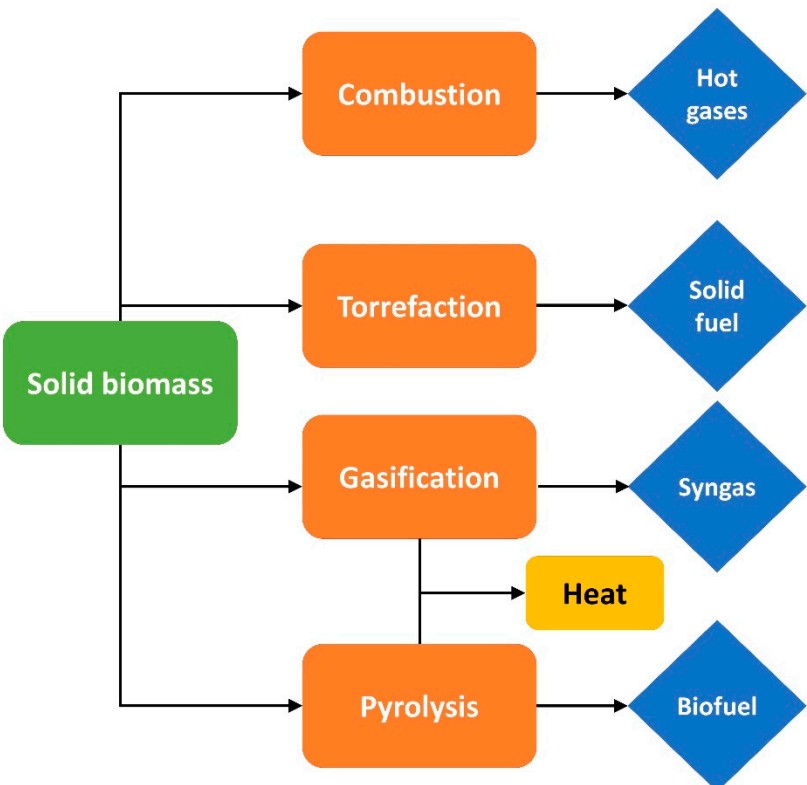

**Figure 4.** Main biomass thermochemical conversion technologies. Own design, based on information from [35–37].

Essentially, all thermochemical conversion technologies are available on a commercial scale, depending on the feedstock in use, although it should be noted that combustion

is more widely applied than other technologies. Examples of commercial facilities that produce each thermochemical pathway are shown in Table 4.

**Table 4.** Examples of commercially available thermochemical technologies.

| Pathway | Reactor Type | Capacity | Developer |
|---------|-------------|----------|-----------|
| Torrefaction | Fluidised bed | 60,000 ton/a | Topell Energy |
| Gasification | Updraft | 2–15 MW | DTI |
| (Fast) Pyrolysis | Fluidised bed | 24,000 ton/a | BTG Bioliquids |

As of 2018, the pulp and paper sector, as well as the wood and wood product industries, used a combined 81% of the biomass used in EU industries for energy consumption. The non-metallic minerals sectors, including glass, ceramics, and cement, are by volume, the third largest industrial users of biomass. Other EII sectors, including the chemical and petrochemical, iron and steel, and non-ferrous metals sectors, use 0.64%, 0.04%, and 0.03% of the biomass for energy consumption, respectively [38].

Biomass combustion to produce heat in combination with electricity is widely applied in several EII sectors. One example is the Polaneic Green Unit in Poland, where the older pulverised coal boiler was replaced with a biomass-fired circulating fluidised bed boiler [39]. Torrefied biomass was applied in the iron and steelmaking industry in existing blast furnaces. Steelmaker ArcelorMittal, Belgium, has started the construction of a new facility called the Torero plant. The produced torrefied biomass will partly substitute pulverised coal and be used as an alternative carbon source [40]. Gasification plants are mainly dedicated to producing heat and electricity from which the heat is used for district heating. There are, however, a few examples of using 'producer gas' for pyro-processing systems in cement plants. One such example is in Germany at the Rüdersdorfer Zement GmbH cement plant [41].

Biochar as a by-product of biomass gasification and pyrolysis is of interest to energy-intensive industries as a substitute for fossil coal used in steel production [42]. Biochar in multiple formats has been tested and compared with anthracite reference coals. Melting tests in a pilot electric arc furnace have shown that biochar reacts in a similar way to reference coals. Thus, biochar shows great potential for use in industrial-scale electric arc furnace steelmaking as a substitute for fossil coal. Additionally, biochar obtained from pyrolysis and gasification can be carbon-negative by combining net carbon removal from industrial processes with the production of energy or other added-value products beyond sequestered carbon [43].

### 3.5. Liquid and Gaseous Biofuels

Biofuels are obtained via the conversion of an organic feedstock either into a liquid (most common), solid, or gaseous form of fuel [44]. Biofuels can be identified depending on the feedstock used for their production in conventional and advanced biofuels [45]. Although conventional biofuels (first-generation biofuels) are known to be produced from edible and land-consuming feedstocks, advanced biofuels (second-generation biofuels) make use of non-food and non-feed organic feedstocks [46]. Although most commercialised biofuels (e.g., biodiesel and bioethanol) are used in the transport sector [47], they are not extensively used in energy-intensive industries within the cement, iron, ceramic, and chemical sectors, to name a few. These sectors still rely on the use of conventional fossil fuels for their processes such as combustion-carbon-based electricity and natural gas for heat production. The former can be substituted more frequently with renewable electricity from variable sources. However, the combustion of natural gas could ideally be substituted by biomethane [48]. Not only is this renewable gas obtained via the anaerobic digestion of multiple renewable organic feedstocks but its use in industries also does not require any modification of current industrial processes. The similarity between the compositions of natural gas and biomethane is very high (Table 5).

**Table 5.** Comparison of natural gas, biogas, and biomethane.

| Compound | Natural Gas (%) [49] | Biogas (%) [50] | Biomethane (%) [51] |
|----------|---------------------|-----------------|---------------------|
| Methane | 87.0–98.0 | 50–75 | >90 |
| Ethane | 1.5–9.0 | N.A. | N.A. |
| Butane | 0.1–1.5 | N.A. | N.A. |
| Pentane | <0.4 | N.A. | N.A. |
| $N_2$ | 5.5 | 0–10 | N.A. |
| CO | 0.05–1.0 | 25–50 | N.A. |
| $O_2$ | <0.1 | 0–2 | N.A. |
| $H_2$ | N.A. | 0–1 | <5 |

Independent of the technical feasibility of biomethane, one of its implementation challenges is deeply linked to its availability [52]. It is expected that biomethane will only replace around 8% of the total natural gas consumption in the EU by 2030 [53].

*3.6. Green Hydrogen*

Hydrogen is an energy carrier that can be produced from fossil fuels and biomass, water, or a mixture of both. At present, roughly 95% of worldwide hydrogen production comes from fossil fuels [54]. Hydrogen is considered renewable or green when the full life-cycle greenhouse gas emissions of the production process are close to zero. The most common method of producing green hydrogen is through the electrolysis of water (in an electrolyser powered by electricity), and with the electricity stemming from renewable sources, it can also be produced through other pathways. FCH JU—The Fuel Cells and Hydrogen Joint Undertaking—investigated 11 different green hydrogen pathways besides electrolysis, and 6 pathways were considered sound for 2030 [55]. Figure 5 presents these different pathways based on the three feedstocks that can be used to generate hydrogen: renewable electricity, biomass and biogas, and solar irradiation. The figure also shows their technological readiness levels (TRL) on the horizontal axis. The steam reforming of biomethane/biogas with or without carbon capture and utilisation/storage is also a mature and well-established technology. Less mature pathways include biomass gasification [56] and pyrolysis [57], thermochemical water splitting [57], photocatalysis [58], the supercritical water gasification of biomass [59], combined dark fermentation [60], and anaerobic digestion.

Currently, there is no significant hydrogen production from renewable sources; green hydrogen has been limited to demonstration projects [54] but is expected to be developed in the coming years. In low- and medium-grade heat industrial segments, using renewable electricity is the primary way to decarbonise industrial processes according to FCH JU [62]. However, electric heaters, boilers, and furnaces become less efficient when higher temperatures are required, and their use may necessitate major adaptations of existing production processes. For industrial processes in the high-grade heat segment, hydrogen may, therefore, offer benefits due to its ability to generate high temperatures using process setups similar to those used today. As more than 30% of the industries' CO2 emissions stem from high-grade heat, these uses have an essential role to play in decarbonisation, certainly for as long as CCS or other innovations are not competitive. Besides its use in high-grade heat processes, EIIs can use green hydrogen for chemical and synthetic fuel production and as a reduction agent (steel industries).

The production of green hydrogen is generally not yet carried out on a commercial scale. It can be used directly for heat and/or power generation, for material use (in the chemical and refinery industries), in production processes to avoid the production of $CO_2$ emissions (in the steel industry), or in CCU processes (in the lime and cement industries). In the steel industry, some companies are exploring the use of hydrogen in their production processes. The three Swedish steel-sector companies, steel manufacturer SSAB, mining

company LKAB, and energy company Vattenfall, collectively work on HYBRIT (Hydrogen Breakthrough Ironmaking Technology) and are developing a pilot plant at the SSAB site in Luleå, Sweden. In Germany, the multinational steel production company ArcelorMittal is taking steps to reduce its carbon emissions by retrofitting a production plant to use hydrogen for iron ore reduction. ArcelorMittal partnered with the University of Freiberg to test a hydrogen-based process at its Hamburg steel production plant [63].

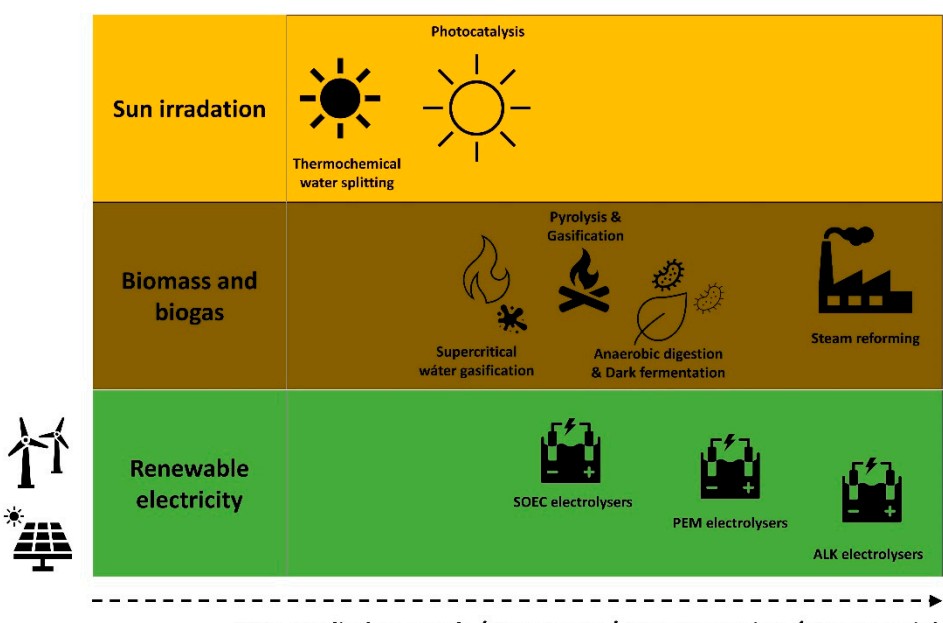

**Figure 5.** Renewable hydrogen pathways and current levels of maturity. TRL stands for technology readiness level (redrawn from the original source [61]).

Regarding green hydrogen, biological hydrogen production via dark fermentation, photosynthesis, or photofermentation could also be considered [64]. Biohydrogen production is of interest to the scientific community and represents an additional route to green hydrogen production [65]. In the current European context, there exist ambitious goals for green hydrogen produced from electrolysers powered with renewable electricity (6 GW by 2024, 40 GW from 2025 to 2030, and green hydrogen deployment on a large scale by 2030). It seems unlikely that biological hydrogen production processes could cope with such goals [66], especially when biological hydrogen production has only been demonstrated at a low TRL [67].

## 4. Assessment of Renewables

Table 6 shows an overview of REs' potential for the decarbonisation of EIIs. The integration of electrified processes is widely applied in industries such as secondary steel and non-ferrous metals production [68]. In high $CO_2$-emitting industries, such as ceramics, glass, and paper, the electrification of the processes will contribute to the reduction in emissions. In other industrial sectors that rely significantly on the use of (fossil-based) heat for the conversion of raw feedstocks, the use of renewable power can be part of the decarbonisation pathway. Renewable power in these highly heat-dependent sectors will have to be used in combination with other renewable solutions.

**Table 6.** Overview of renewable energies' potential for the decarbonisation of the top GHG-emitting and energy-intensive industries.

| Sector | Renewable Power for Process Electrification | | Renewable Heat and Its Sources | | | | CCUS Technologies | |
|---|---|---|---|---|---|---|---|---|
| | Heat and Mechanical | Electrochem. Processes (Excl. $H_2$) | Biomass Combustion (and Biofuels Feedstock) | Other RE (Geotherm. and Conc. Solar) | Green $H_2$ (Electrolysis/Gasification) | Biomethane (Anaerobic Digestion) | Carbon Capture and Storage | Carbon Capture and Utilisation |
| Steel | XXX | XX | X | XXX | XXX | XXX | XXX | XXX |
| Chemicals | XXX | XXX | XXX | XX | XXX | XX (**) | XXX (*) | XXX |
| Fertilisers | XXX | XXX | XXX | XX | XXX | XX (**) | XXX (*) | XXX |
| Cement | XX | O | XXX | XX | X | XX (**) | XXX | XXX |
| Lime | X | O | XXX | XX | X | XXX | XXX | XXX |
| Refining | XX | O | XXX | XX | XXX | XXX (**) | XXX | XXX |
| Ceramics | XXX | O | X | XXX | XX | XXX | O | X |
| Paper | XX | O | XXX | XX | O | XXX (**) | O | O |
| Glass | XXX | O | XXX | XXX | X | XXX | O | O |
| Non-Fe metals | XXX | XXX | XXX | XX | XX | XXX | X | X |
| Alloys | XXX | XXX | XXX | XX | XX | XXX | X | X |
| Notes | O: Limited or no significant application foreseen<br>X: Possible application but no main route or wide-scale applications<br>XX: Medium potential<br>XXX: High potential | | | | XXX: Sector already applies the technology on a large scale (it can be expanded in some cases)<br>(*) In particular for ammonia and ethylene oxide<br>(**) In particular as feedstock for added-value products | | | |

Table adapted from the original source: [69].

One of the main opportunities for renewable power in sectors where heat is required, especially low-temperature heat, will be the use of electric boilers capable of providing heat application below 300 °C. Industrial applications, such as electric arc, infrared, induction, dielectric, direct resistance, microwave, and electron beam heating, which require temperatures below 1000 °C, can be powered with renewable electricity [69]. Above 1000 °C, there is a need for further research to fulfil the requirements of industrial sectors such as the cement and glass sector [68].

Apart from the electrification of heat with renewable power, this will be widely employed in conventional electrochemistry-based technologies [70] and novel electrochemistry-based fermentations for high-added value products from $CO_2$ [71]. The non-ferrous, ferro-alloys and silicon, and chemical industries make use of these technologies that will continue to be powered with electricity. However, there will be a gradual increase in the use of renewable power to replace fossil-based power. Examples of the use of renewable power in the steel and chemical industries are high-temperature steel electrolysis/iron ore reduction with plasma and electricity-based processes (plasma, microwave, and ultrasounds).

Multiple and diverse industrial processes in the steelmaking and refining sector (e.g., feedstock for ammonia production) can utilise (renewable) hydrogen. Today, most global $H_2$ production relies on the steam reforming of methane (SMR), which is a process that leads to high $CO_2$ emissions [72]. Thus, low $CO_2$-emitting routes such as electrolysis are attracting attention worldwide. However, other alternative routes, such as methane pyrolysis [73], water photolysis, and standard SMR with CCSU, should also be considered.

Applications of hydrogen obtained using the above-mentioned routes will continue to be for the refining of chemicals (for low-$CO_2$ ammonia, methanol, olefins, or (bio)synthetic fuel production [74]) and in the steel sector (as a reducing agent in the direct reduction of iron or in smelting processes) [75].

Biomass (wood, agricultural, and forestry waste) will be a key feedstock in the decarbonisation of industries independent of its origin [76]. Biomass is used and will continue to be used not only as a fuel for heat production in the paper and silicon industries but

also as a feedstock for other types of biofuels. For example, biomass is used as a partial replacement for coal as a reducing agent in the steelmaking industry [40]. In recent years, lignocellulosic ethanol production technologies have reached the commercialisation stage [77]. Furthermore, algae biomass has successfully been used for the production of biofuels such as biogas [78]. An issue regarding the use of biomass, either as a direct source of heat via combustion or as a feedstock for biofuels, is its availability. Although worldwide biomass is the largest renewable energy source with a significant 13% share of the global energy mix (considering all types of energy sources) [79], it is estimated that only half of both the energetic and non-energetic final consumption of biomass by the chemical sector can be covered [69].

Renewable gases such as hydrogen and biomethane will become increasingly relevant for the replacement of natural gas [80]. Biomethane as the result of biogas upgrading from the anaerobic digestion of organic matter and via thermochemical methanation processes is considered a renewable gas that can be injected into the gas grid and used to substitute fossil-based natural gas without the need to adapt or change equipment. Green hydrogen produced via renewable power requires replacement of the burners and part of the gas installation to ensure connection tightness. However, biomass-based solutions face challenges with respect to availability. It is expected that biomethane will only replace around 8% of the total natural gas consumption in the EU by 2030. On the other hand, only 40 GW of green $H_2$ will be available in the EU by 2030 [81].

Finally, carbon capture use and storage (CCUS) will bring further opportunities to industries for decarbonising their processes. Currently, 40 Mt $CO_2$ is captured each year in industries and fuel transformation [82]. There exists a plethora of robust $CO_2$ capture technologies that are applied in pre- and post-combustion processes in industries [83]. Once $CO_2$ has been captured, the main route should be its valorisation in added-value products, for example, in the fertiliser and chemical industries. In the latter, $CO_2$ as a feedstock can be used to produce a broad range of products from basic chemicals to fine chemicals and synthetic fuels [69]. Here, the intervention of renewable $H_2$ will be essential. Interest in the use of $CO_2$ has increased in different industrial sectors. Specifically, $CO_2$ contained in waste gases coming from iron and steel production can be used to produce ethanol thanks to the action of specialised microbial cultures [84]. In addition to the microbial production of ethanol from $CO_2$, multiple compounds could potentially be produced from industrial $CO_2$ such as acetate, methanol, and butyrate, among others [85,86]. These simple compounds could lead to multiple value chains thanks to additional post-processes to convert carboxylates into bulk fuels or solvents [87].

Carbon capture and storage (CCS) will be an alternative mitigation route for large industrial $CO_2$ emitters such as the cement and lime, steel, chemical manufacturing, and refining sectors [88]. Highly $CO_2$-concentrated gas streams can be easily processed since these streams do not require the technologies that currently capture $CO_2$ from diluted streams. Examples of this are the exhaust streams found in ammonia production via the steam methane reforming process (95–100% $CO_2$) and in ethylene production (30–100% $CO_2$) [69].

An industrial example of carbon capture bioconversion into synthetic fuels is currently being developed by ArcelorMittal. Carbon from the blast furnace is captured and taken to a bioreactor. There, the gas is pressurised and injected into a microbial broth where microorganisms consume CO and $H_2$ to produce ethanol, which is continuously distilled from the broth [89]. Some characteristics that make this technology relevant for the decarbonisation of steel production are (i) the microorganisms used in this process are driven by CO and $H_2$; in case of a lack of $H_2$, the microorganisms can perform the water shift reaction by "picking up" $H_2$ from the water; (ii) there is no need for a strict cleaning of the blast furnace gas since the microorganisms can easily adapt to trace impurities such as $CO_2$ and nitrogen; (iii) thanks to the use of different specialised microbial cultures, the production of multiple chemicals can be accomplished; and (iv) the energy conservation in the system

has been hypothesised to be relevant; around 70% could be converted into alcohol and approximately 30% could be converted into biomass that could be further valorised.

## 5. Conclusions

EIIs' decarbonisation will occur through the progressive use of an energy mix that allows the EU industrial sectors to remain competitive on a global scale. Each industrial sector will require specific renewable energy solutions, especially the top GHG-emitting industries. This work provides a catalogue of renewable energy technologies that are currently available (regarding the 2030 scope) and that will be available in the transition towards 2050. These renewable options have been classified into technologies based on the use of renewable electricity and those used in industries to produce heat for multiple processes. Process electrification will be the key to the decarbonisation of industries. Although the costs of producing renewable electricity will gradually decrease, current natural gas-dependent processes will need to be converted to take advantage of the increasing availability of renewable power. Industrial processes that are not viable for electrification will still need a form of renewable heat. Multiple renewable technologies are and will become available in the coming years to supply a broad range of temperatures needed by industries ranging from concentrating solar power and heat pumps to geothermal energy. Biomass will be a key element in decarbonisation not only in coal-based combustion systems but also as a feedstock for biofuels. The contribution of renewable gases such as biomethane from anaerobic digestion and green hydrogen from electrolysis is expected to be essential in the coming years. Biomethane could allow a "smooth" transition due to its chemical composition and remarkable renewable origin. Green hydrogen will not only require technological adaptations but also several years to increase its availability all over Europe. The present work also serves as an initial point of discussion with potential decision makers to jointly draft a vision of renewable energy solutions for the development of short- and long-term strategies for the full decarbonisation of EIIs.

**Author Contributions:** Conceptualisation, A.A.C.-M., R.J., P.R., E.K. and C.J.-C.; Writing—original draft preparation, A.A.C.-M., A.F.-C., C.K., R.J., B.D., P.R. and M.V.; Writing—review and editing, A.A.C.-M., A.F.-C., A.R., O.B., R.J., B.D., P.R., M.V., E.K. and C.J.-C.; Supervision, A.A.C.-M., A.R. and C.J.-C.; Project administration, A.R.; Funding acquisition, C.K., P.R., E.K., P.G. and C.J.-C. All authors have read and agreed to the published version of the manuscript.

**Funding:** The RE4Industry project has received funding from the European Union's Horizon 2020 research and innovation programme under grant agreement No. 952936.

**Acknowledgments:** A.C. acknowledges DALL-E 2 for generating digital images for the graphical abstract.

**Conflicts of Interest:** The authors declare no conflict of interest.

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
