# Peer review of "Renewable Power and Heat for the Decarbonisation of Energy-Intensive Industries"

_processes, doi:10.3390/pr11010018_

Round 1

Reviewer 1 Report

The manuscript entitled “ Renewable power and heat for the decarbonisation of energy intensive industries” is  a very comprehensive review on this topic. I appreciate the authors for such a nice collaboration in bringing high valued review article. In my opinion, the review article merits publication in Processes.

However, few suggested were made herewith. Inclusion of responses to the manuscript will improve the quality.

1.     Many instances the relevant references were missing. Include them. It enables for clear cross referring the manuscript contents.

2.     Revise the sentence: Line 85: This shows the ….. of energy intensive industries might experience a similar cost declining trend.

3.     “Table 1. Renewable electricity-based technologies for the electrification of industrial.” Revise. Incomplete title.

4.     Along with ‘district heating’, use the synonyms of it. The term ‘district’ got different meaning in various geographical locations of the globe.

5.     Line 156: Change “Air source pumps” to “Air source heat pumps”

6.     Line 178: Revise the sentence: provides a representative overview … applications in several industrial sectors[25].

7.     Line 222: change “physical-chemical” with correct phrase

8.     In Figure 4: The connection of boxes in grey colour with the biomass energy is not clear. Revise.

9.     Line 289: Which figure is representing TRLs?

10.  Under green energy there is high scope in future with biohydrogen from the treatment of industrial and food-based wastewaters. Authors may refer and cite the following articles regarding biohydrogen. https://doi.org/10.1016/j.apenergy.2013.01.085; https://doi.org/10.1016/j.ijhydene.2011.10.109.

11.  Revise Line 393: Specifically, CO2 contained in waste gases …. thanks to the action of specialized microbial cultures[80].

12.  Along with ethanol, many other products can be produced from CO2 through microbial action. Authors can consider the following articles. https://doi.org/10.1039/C7RE00220C; https://doi.org/10.1016/j.jcou.2016.03.003

13.  In figure 5: It looks something is missing. An improvement with clarity is recommended.

Author Response

Please note that a PDF has been uploaded with responses to reviewers

The manuscript entitled “Renewable power and heat for the decarbonisation of energy intensive industries” is a very comprehensive review on this topic. I appreciate the authors for such a nice collaboration in bringing high valued review article. In my opinion, the review article merits publication in Processes.

However, few suggested were made herewith. Inclusion of responses to the manuscript will improve the quality.

First of all, we thank the reviewer for her/his valuable comments that have helped us improve the overall quality of the manuscript and to clarify the gaps in information detected by her/him.

  1. Many instances the relevant references were missing. Include them. It enables for clear cross referring the manuscript contents.

We agree with the reviewer that the proper inclusion of references enables for clear cross referring the manuscript contents. Reviewer 1 points out an issue regarding the way references appear in the document. Instead of showing the number of the citation, it says: “Error! Reference source not found.”:

There might have been a problem while converting the word document to PDF. In the current word version of the manuscript, all references are properly shown with the required format by MDPI.

  1. Revise the sentence: Line 85: This shows the ….. of energy intensive industries might experience a similar cost declining trend.

The following sentence has been revised:

This shows the competitiveness of renewable power generation and that process electrification required in the decarbonisation of energy intensive industries might experience a similar cost declining trend.

We are not sure what to revise in the present sentence. In the absence of a specific comment by Reviewer 1, we would like to inform her/him and eth editor that the sentence will remain in its current form.

  1. “Table 1. Renewable electricity-based technologies for the electrification of industrial. Revise. Incomplete title.

The title of Table 1 has been revised and completed as follows.

Previous title: Table 1. Renewable electricity-based technologies for the electrification of industrial.

Current tile: Table 1. Renewable electricity-based technologies for the electrification of industrial processes.

  1. Along with ‘district heating’, use the synonyms of it. The term ‘district’ got different meaning in various geographical locations of the globe.

It is not clear what the suggestion by the reviewer is. Can he/she please specify?

  1. Line 156: Change “Air source pumps” to “Air source heat pumps”

Thanks. The change has been made.

  1. Line 178: Revise the sentence: provides a representative overview … applications in several industrial sectors[25].

The sentence has been revised and modified as follows.

Previous: Table 3 provides a representative overview of the potential of heat pump applications in several industrial sectors.

Current: Table 3 provides an overview of the potential of heat pump applications in several industrial sectors.

  1. Line 222: change “physical-chemical” with correct phrase

The sentence has been revised and modified as follows.

Previous: Thermo-chemical, physical-chemical, and bio-chemical.

Current: Thermochemical, physicochemical, and biochemical.

  1. In Figure 4: The connection of boxes in grey colour with the biomass energy is not clear. Revise.

Thanks to this comment by Reviewer 1 we have noticed that grey boxes in Figure 4 do not properly deliver the message that conversion processes of solid biomass might give as a result either heat or electricity for industrial processes.

Taking this into consideration we have removed those grey boxes from Fig. 4 to avoid confusion:

  1. Line 289: Which figure is representing TRLs?

We agree with the Reviewer that it is not clear where the technology readiness level of each technology depicted in Fig. 5 can be identified. We have included the acronym “TRL” in the figure in the bottom part and a brief explanation in the text as follows:

Figure 5 presents these different pathways based on the three feedstock that can be used to generate hydrogen: Renewable electricity, Biomass and biogas and Solar irradiation. The figure also shows their technological readiness level (TRL) in the horizontal axis.

Figure 5. Renewable hydrogen pathways and current levels of maturity. TRL stands for technology readiness level (Redrawn from the original source[61]).

  1. Under green energy there is high scope in future with biohydrogen from the treatment of industrial and food-based wastewaters. Authors may refer and cite the following articles regarding biohydrogen.

https://doi.org/10.1016/j.apenergy.2013.01.085;

https://doi.org/10.1016/j.ijhydene.2011.10.109.

We agree with the Reviewer that biohydrogen production (in its multiple routes0F[1]) is of interest amongst researchers in the scientific community. Thus, it is worth mentioning these biological hydrogen production processes as additional routes for green hydrogen production. Specially the suggested articles.

However, in the current context of green hydrogen production set by governments, specially in Europe, with very ambitious goals in 2024 (6 GW), 2025-2030 (40 GW) and 2030 and beyond (deployed at large scale)1F[2], it seems difficult to see biological hydrogen production processes compete with green hydrogen production from electrolysers powered with renewable electricity.

We suggest the following addition to the green hydrogen section:

Regarding green hydrogen, biological hydrogen production via Dark fermentation, Photosynthesis or Photofermentation could also be considered [Ref.2F[3]]. Biohydrogen production is of interest in the scientific community, and it represents an additional route for green hydrogen production [Ref.3F[4]]. In the current European context, there exists ambitious goals of green hydrogen being produced from electrolysers powered with renewable electricity (6 GW in 2024, 40 GW from 2025 to 2030 and green hydrogen deployment at large scale by 2030). It seems unlikely that biological hydrogen production processes could cope with such planned goals2, especially when biological hydrogen production has only been demonstrated at a low TRL4F[5].

  1. Revise Line 393: Specifically, CO2 contained in waste gases …. thanks to the action of specialized microbial cultures[80].

In this comment by the Reviewer we believe that he/she refers to his/her next comment regarding the multiple products that can be obtained from converting CO2 via the action of microbial cultures.

  1. Along with ethanol, many other products can be produced from CO2 through microbial action. Authors can consider the following articles.

https://doi.org/10.1039/C7RE00220C; https://doi.org/10.1016/j.jcou.2016.03.003

We agree with the reviewer that multiple products can be obtained from CO2 through the action of microorganisms. We have tried now to show such variety while at the same time citing the suggested references:

In addition to the microbial production of ethanol from CO2, there are multiple compounds that could potentially be produced from industrial CO2: acetate, methanol, butyrate, among others [Refs.5F[6]]. These simple compounds could lead to multiple value chains thanks to additional post-processes to convert carboxylates into bulk fuels or solvents6F[7].

  1. In figure 5: It looks something is missing. An improvement with clarity is recommended.

We have tried to improved the clarity of Fig 5. Please see both versions as follows. Previous version and the improved version are presented here but only the improved version is now included in the main manuscript:

Previous version

Current improved version

[1] Dark fermentation, Photosynthesis, Photofermentation, among others.

[2] EU Commission. A Hydrogen Strategy for a Climate Neutral Europe. 2020.

[3] https://doi.org/10.1016/j.apenergy.2013.01.085

[4] https://doi.org/10.1016/j.ijhydene.2011.10.109

[5] https://doi.org/10.1007/s13399-020-00965-x

[6] https://doi.org/10.1039/C7RE00220C & https://doi.org/10.1016/j.jcou.2016.03.003

[7] Agler, M. T., Wrenn, B. A., Zinder, S. H., & Angenent, L. T. (2011). Waste to bioproduct conversion with undefined mixed cultures: the carboxylate platform. Trends in biotechnology29(2), 70-78.

Reviewer 2 Report

Dear Authors,

thank you for the work you have done. The document is of great interest, well organized and well written.

However, I must point out that the treatment of carbon-capture and storage through the char produced by biomass gasification and pyrolysis is missing. It is a topic of fundamental importance and in great development in recent years.

I therefore ask you to add this discussion to paragraph 3.4 and chapter 4.

BECCS systems are one of the most effective strategies to date for the removal of atmospheric CO2, they are worth discussing in depth. In the PDF you will find this and other (minor) comments.

Best regards,

NM

Author Response

Please note that a PDF has been uploaded with responses to reviewers

Dear Authors,

thank you for the work you have done. The document is of great interest, well organized and well written.

However, I must point out that the treatment of carbon-capture and storage through the char produced by biomass gasification and pyrolysis is missing. It is a topic of fundamental importance and in great development in recent years.

I therefore ask you to add this discussion to paragraph 3.4 and chapter 4.

BECCS systems are one of the most effective strategies to date for the removal of atmospheric CO2, they are worth discussing in depth. In the PDF you will find this and other (minor) comments.

Best regards,

NM.

Comments made by Reviewer 2 found in the PDF version of the manuscript:

Specific comment 1 - Page 2, Line 46: “Check this reference please”. Reviewer 2 points out an issue regarding the way references appear in the document. Instead of showing the number of the citation, it says: “Error! Reference source not found.”:

There might have been a problem while converting the word document to PDF. In the current word version of the manuscript, all references are properly shown with the required format by MDPI.

Specific comment 2 – Figure 2: “should this line be red instead of orange? the same for the blue line below. check the graph on the right too please”. Reviewer 2 indicates that a correction to be made in Figure 2 to homogenize format as follows:

We have corrected this issue to homogenize the format of Figure 2 and included the following version in the latest version of the manuscript:

Specific comment 3 - Page 3, Line 98: “i would say power, not energy. my opinion”. In the following sentence, Reviewer 2 has made the suggestion to include the term power instead of energy:

“Current worldwide photovoltaic energy capacity is expected to grow from 900 GW (EU share of 25%) installed to 3000 GW (EU share of 99 5%) installed by 2050 according to the IEA Roadmap[7].”

We agree with Reviewer 2. The sentence has been modified in the manuscript as follows:

Current worldwide photovoltaic power capacity is expected to grow from 900 GW (EU share of 25%) installed to 3000 GW (EU share of 99 5%) installed by 2050 according to the IEA Roadmap [7].

Specific comment 4 - Page 11, Line 254: “In this chapter an important part is missing: gasification and pyrolysis produce char as byproduct. This material can shift the process towards carbon-negativity. In fact, this is one of the few carbon-capture and storage (CCS and BECCS) process nowaday available. I think it is worth to mention this.”.

We thank Reviewer 2 for pointing out the importance of char produced in gasification and pyrolysis. We do agree that char has not been sufficiently discussed in the present manuscript. For the sake of completeness, we have included the following information at the end of Section “3.4 Solid biomass”:

Biochar as a by-product of biomass gasification and pyrolysis is of interest for energy intensive industries as substitutes for fossil coal used in the steel production7F[1]. Biochar in multiple formats has been tested and compared to anthracite reference coals. Melting tests in a pilot electric arc furnace have shown that biochar reacts in a similar way as reference coals. Thus, biochar shows a great potential for its use in industrial scale electric arc furnace steel making as a substitute of fossil coal. Additionally, biochar obtained from pyrolysis and gasification can be carbon negative by combining net carbon removal from industrial processes with the production of energy or other added-value products beyond sequestered carbon8F[2].

We also thank Reviewer 2 for commenting on Bioenergy with carbon capture and storage (BECCS) systems. They are indeed very promising solutions to produce renewable energy with a neutral or even a negative carbon footprint. Within the scope of the present review we have focused on technologies that might be of relevance for the decarbonisation of energy intensive industries in Europe. BECCS systems per se might be out of the scope of the present review since energy intensive industries might focused instead on the production of synthetic fuels from their own process emissions.

An example of this is the EU-funded Torero project by ArcelorMittal9F[3]. They will produce bioethanol from the microbial fermentation of captured carbon dioxide coming from a blast furnace. Although other decarbonisation projects by this company are briefly discussed in the manuscript, there is no mention to their plans to produce synthetic bioethanol from captured CO2.

We have included the following paragraph regarding this CO2 bioconversion project by ArcelorMittal in section 4 where we discuss the relevance of synthetic fuels from captured industrial CO2:

An industrial example of carbon capture bioconversion into synthetic fuels is currently being developed by ArcelorMittal. Carbon from the blast furnace is captured and taken to a bioreactor. There, the gas is pressurized and injected into a microbial broth where microorganisms consume CO and H2 to produce ethanol which is continuously distilled out of the broth10F[4]. Some characteristics that make this technology relevant for the decarbonisation of steel production are: i) microorganisms used in this process are driven by CO and H2. In case of lack of H2, it is expected that microorganisms might be able to perform the water shift reaction by “picking up” H2 from water; ii) there is no need of a strict cleaning of the blast furnace gas since microorganisms can easily adapt to trace impurities, CO2 and nitrogen; iii) thanks to the use of different specialized microbial cultures, the production of multiple chemicals can be accomplished; and iv) the energy conservation in the system has been hypothesized relevant, around 70% could be converted into alcohol, while approximately 30% can be converted to biomass that could be further valorised.

Specific comment 5 - Page 14, Line 370: “Also in this paragraph it is important to stress the capability of gasification/pyrolysis based unit to be carbon-negative. Look for BECCS systems for reference.”.

We have already addressed these two comments by Reviewer 2 regarding the capability of gasification/pyrolysis based units to be carbon-negative and Bioenergy with carbon capture and storage (BECCS) systems. Please see above.

[1] Demus, T., Reichel, T., Schulten, M., Echterhof, T., & Pfeifer, H. (2016). Increasing the sustainability of steel production in the electric arc furnace by substituting fossil coal with biochar agglomerates. Ironmaking & Steelmaking43(8), 564-570.

[2] Brown, R.C. The Role of Pyrolysis and Gasification in a Carbon Negative Economy. Processes 20219, 882. https://doi.org/10.3390/pr9050882

[3] http://www.torero.eu/

[4] De Maré, C. (2021). Why Both Hydrogen and Carbon Are Key for Net-Zero Steelmaking. IRON & STEEL TECHNOLOGY1.

Author Response

Please note that a PDF has been uploaded with responses to reviewers

Decarbonization is an important topic. The manuscript seems to talk about the decarbonization options for energy intensive industries, i.e., paper and pulp, chemical, minerals..., in view of renewable electricity, and heat. However, I am quite confused when going through the manuscript because the manuscript was not well organized. Details below...

First, we thank Reviewer 3 for his/her comments. We regret that our manuscript might not seem to be well organized for Reviewer 3. This could have been caused by the extension given to discuss the forms of Renewable heat in Section 3 (7 pages) when compared to Section 2 (2 pages) focused on Renewable electricity.

The main reason to give a shorter length in the manuscript to Renewable electricity is that most of the Energy Intensive Industries nowadays acquire renewable electricity through power purchase agreements. Thus, it is not common for a steel, cement or a glass factory to produce its own electricity on site. The focus of these energy intensive industries in Europe seems to be the electrification of their processes as much as possible.

That is why we have focused our analysis regarding renewable electricity on price reduction and industrial process electrification.

  1. In the section of “Renewable Electricity”, it looks like the authors want to talk about the potential resources that can be converted into renewable electricity. However, only solar, and wind electricity was discussed a bit while hydropower, geothermal energy, and biomass represent some other important renewable energies.

We thank this comment by Reviewer 3. In Section 2 “Renewable electricity”, we have focused our analysis on pointing out the following two facts instead of discussing in detail multiple options to produce renewable electricity: i) the competitive decreasing prices of renewable electricity production from photovoltaics, concentrating solar power and off-/on-shore wind (Fig. 1) and ii) the electrification of processes that currently run on fossil fuels like natural gas or coal as detailed in Table 1.

Other technologies to produce renewable heat, like Geothermal energy, Heat pumps, Solid biomass, Liquid-gaseous biofuels and green Hydrogen have been covered in Section 3 “Renewable heat”. We agree with Reviewer 3 that there exist technologies to produce renewable heat for energy intensive industries. Thus, a whole section has been focused on discussing how these can be employed by energy intensive industries for the decarbonisation of their processes. Please see Section 3.

  1. In the sections of “Renewable heat”, the authors seem like trying to talk about both the resources and technologies that are applied to generate renewable heat. But, no offense, I think here the text is messing things up. I would suggest to talk about the raw resources, like wind, biomass, the refined energies, like green hydrogen, biofuel, and the renewable energy technologies, like heat pump, in different sections, so that readers will not get lost.

We thank this comment by Reviewer 3. However, we would like to emphasize that what Reviewer 3 is suggesting has already been done in the initial version of the manuscript as a structure of Section 3:

  1. Concentrating solar power;
  2. Heat pumps;
  3. Geothermal energy;
  4. Solid biomass;
  5. Liquid and gaseous biofuels; and
  6. Green hydrogen.

Renewable energy solutions have been classified into technologies based on the use of renewable electricity (Section 2) and those to be used to produce heat for multiple industrial processes (Section 3).

Industrial processes that are not readily eligible for using renewable electricity will still be needing a form of renewable heat. In Section 3, we have discussed Concentrating solar power, Heat pumps, and Geothermal energy to supply a broad range of temperatures needed.

Additionally, Solid biomass has been discussed as a key element in the decarbonisation of not only conventional combustion systems but also as feedstock for the production of liquid and gaseous biofuels. Last, the technology readiness level of green hydrogen production technologies is discussed.

  1. Continue to point 2, it also better to talk about CO2 conversion in a separated section.

CO2 conversion into synthetic fuels it is a very promising approach not only to reduce emission by energy intensive industries but to make use of such emissions in the production of synthetic (bio)fuels. This is actually, an approach considered by industries in the steel sector like ArcelorMittal.

Carbon from the blast furnace is captured and taken to a bioreactor. There, the gas is pressurized and injected into a microbial broth where microorganisms consume CO and H2 to produce ethanol which is continuously distilled out of the broth.

We have included a paragraph regarding this CO2 bioconversion project by ArcelorMittal in section 4 where we discuss the relevance of synthetic fuels from captured industrial CO2:

An industrial example of carbon capture bioconversion into synthetic fuels is currently being developed by ArcelorMittal. Carbon from the blast furnace is captured and taken to a bioreactor. There, the gas is pressurized and injected into a microbial broth where microorganisms consume CO and H2 to produce ethanol which is continuously distilled out of the broth11F[1]. Some characteristics that make this technology relevant for the decarbonisation of steel production are: i) microorganisms used in this process are driven by CO and H2. In case of lack of H2, it is expected that microorganisms might be able to perform the water shift reaction by “picking up” H2 from the water; ii) there is no need of a strict cleaning of the blast furnace gas since microorganisms can easily adapt to trace impurities, CO2 and nitrogen; iii) thanks to the use of different specialized microbial cultures, the production of multiple chemicals can be accomplished; and iv) the energy conservation in the system has been hypothesized relevant, around 70% could be converted into alcohol, while approximately 30% can be converted to biomass that could be further valorised.

We will also cover in detail in a separate future work synthetic fuels production from captured industrial CO2. 

  1. In the sections of “Assessment of Renewables”. It is expected that the benefits of different renewable energy are determined quantitatively.

We understand Reviewer 3’s comment that a quantitatively assessment of renewable energy technologies for the decarbonisation of energy intensive industries could be expected. A quantitatively assessment of renewables is currently being conducted for three study cases of industries within the steel, chemicals and aluminium sector12F[2]. The use of Biomethane and Green hydrogen is included in such analysis. A publication is expected to be released with the case study highlights in renewable energy adoption for decarbonisation of steel, chemicals and aluminium specific sector processes. Unfortunately, at the time of the current submission, a quantitative assessment cannot be made publicly available.

[1] De Maré, C. (2021). Why Both Hydrogen and Carbon Are Key for Net-Zero Steelmaking. IRON & STEEL TECHNOLOGY1.

[2] https://re4industry.eu/2021/10/28/re4-industry-workshops-on-industrial-needs-and-solutions/

Round 2

Reviewer 2 Report

Dear Authors,

The paper has been improved and I consider it can now be published.

Reviewer 3 Report

The manuscript was revised accordingly.